# Restoration of Degraded Tropical Peatland in Indonesia: A Review

Tri Wira Yuwati [1,*], Dony Rachmanadi [1], Pratiwi [2], Maman Turjaman [2], Yonky Indrajaya [3], Hunggul Yudono Setio Hadi Nugroho [3], Muhammad Abdul Qirom [1], Budi Hadi Narendra [2], Bondan Winarno [2], Sri Lestari [2], Purwanto Budi Santosa [1], Rahardyan Nugroho Adi [3], Endang Savitri [3], Pamungkas Buana Putra [3], Reni Setyo Wahyuningtyas [1], Retno Prayudyaningsih [4], Wawan Halwany [1], Besri Nasrul [5], Bastoni [6] and Daniel Mendham [7]

1    Environment and Forestry Research and Development Institute of Banjarbaru, Jl. Ahmad Yani Km. 28.7, Banjarbaru 70721, Indonesia; dony.research@gmail.com (D.R.); qirom.litbanglhk@gmail.com (M.A.Q.); pur.befordi@gmail.com (P.B.S.); renisetyowahyuningtyas@gmail.com (R.S.W.); wawanh73@gmail.com (W.H.)
2    Forest Research and Development Center, The Ministry of Environment and Forestry, Jl. Gunung Batu No. 5, Bogor 16118, Indonesia; pratiwi.lala@yahoo.com (P.); turjaman@gmail.com (M.T.); narendra17511@gmail.com (B.H.N.); bondanw2308@gmail.com (B.W.); lestari@iuj.ac.jp (S.L.)
3    Watershed Management Technology Center, Jl. Ahmad Yani, Surakarta 57102, Indonesia; yonky_indrajaya@yahoo.com (Y.I.); hunggulys@yahoo.com (H.Y.S.H.N.); dd11lb@yahoo.com (R.N.A.); savitriendang@gmail.com (E.S.); pamungkas_buanaputra@yahoo.co.id (P.B.P.)
4    Environment and Forestry Research and Development Institute of Makassar, Jl. P. Kemerdekaan Km. 16, Makassar 90243, Indonesia; rprayudyaningsih@gmail.com
5    Study Program of Soil Science, Faculty of Agriculture, Universitas Riau, Jl. Bina Widya Simpang Baru Pekanbaru, Pekanbaru 28293, Indonesia; besrinasrul@lecturer.unri.ac.id
6    Environment and Forestry Research and Development Institute of Palembang, Jl. Kol. H. Burlian Km. 6.5, Palembang 30153, Indonesia; bastonibrata@yahoo.co.id
7    CSIRO Land and Water, Private Bag 12, Hobart, Tasmania 7001, Australia; Daniel.Mendham@csiro.au
*    Correspondence: yuwatitriwira@gmail.com; Tel.: +62-81-2505-3180

**Abstract:** Tropical peatlands are fragile ecosystems with an important role in conserving biodiversity, water quality and availability, preventing floods, soil intrusion, erosion and sedimentation, and providing a livelihood for people. However, due to illegal logging, fire and conversion into other land use, the peatlands in Indonesia are under serious threat. Efforts to restore Indonesia's tropical peatlands have been accelerated by the establishment of the Peatland Restoration Agency in early 2016. The restoration action policy includes the rewetting, revegetation and revitalisation of local livelihood (known as the 3Rs). This paper summarises the regulatory, institutional and planning aspects of peatland restoration, in addition to the implementation of the 3Rs in Indonesia, including failures, success stories, and the criteria and indicators for the success of peatland restoration.

**Keywords:** rewetting; revegetation; revitalisation; rehabilitation; tropical peatland

## 1. Introduction

The world's peatland ecosystem covers an area of 398 Mha, about 34–45 Mha of which is in tropical climates, with about 56.6% (25 Mha) in South East Asia [1–3]. In Indonesia, peatland covers an area of about 13.4–14.9 Mha [4,5], mainly in Sumatra (5.85 Mha), Kalimantan (4.54 Mha), Papua (3.01 Mha) and Sulawesi (0.03 Mha) [5], with the main distribution seen in Figure 1. From Figure 1 it can be seen that the distribution of tropical peatlands in Indonesia is quite wide. In addition to the enormous benefits, the vast distribution of peat, if mismanaged, will cause disasters such as peatland fires.

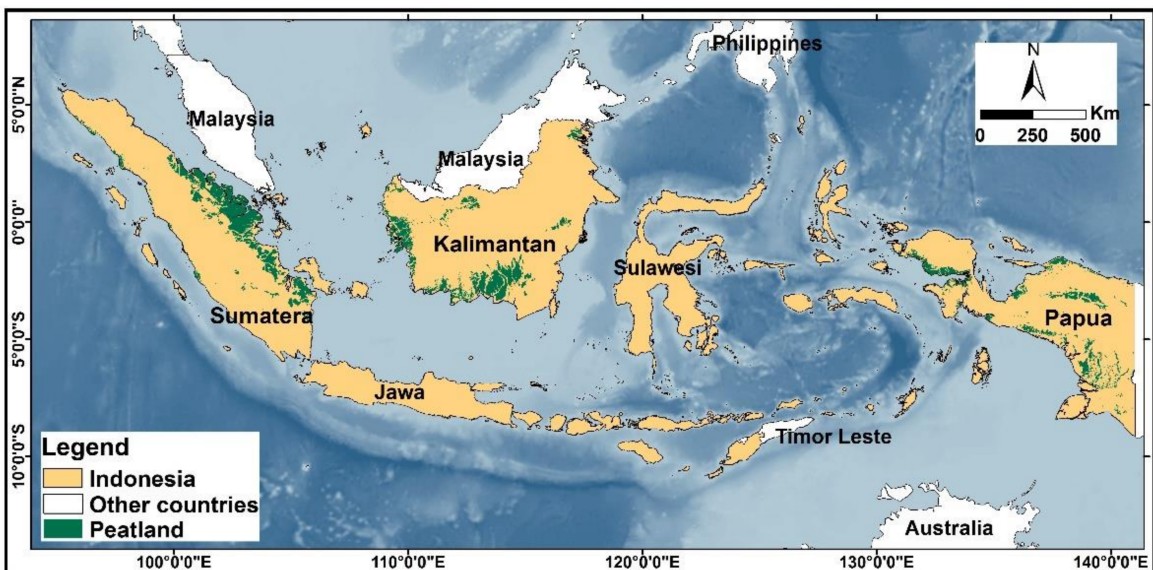

**Figure 1.** Main peatland distribution in Indonesia, in the four big islands of Sumatera (43.5%), Kalimantan (33.8%), Papua (22.4%) and Sulawesi (0.3%).

Although temperate and arboreal peat is typically formed from the remains of aquatic plants (spaghnum), tropical peat is formed from the residue of woody plant material which has a high lignin and cellulose content [6–8]. Tropical peat is formed in two types of systems, classified as either topogenous and ombrogenous [9]. Topogenous peat is formed inland from coastal plains/rivers and is affected by tidal flows from the sea. Ombrogenous peat is formed only from rain water and/or organic matter sourced in the local environment, and has a much lower fertility because of the lack of external mineral input. Topogenous peat is usually located downstream or at lower elevation than ombrogenous peat [9,10].

Based on the level of maturity, tropical peat is divided into three categories, namely: sapric (highly mature), hemic (semimature), and fibric (relatively fresh) [11,12]. The maturity level of peat is strongly influenced by the rate of decomposition so that, in general, the surface layer is relatively more mature. By comparison, in deep peat, layers of peat maturity levels that vary greatly are interpreted as the stages of peat formation time.

Peat of higher maturity/greater decomposition tends to have a greater ability to absorb and bind water [12,13]. On a volumetric basis, peat soil has a water holding capacity of around 0.8–0.9 $m^3/m^3$, and on a gravimetric basis, the water holding capacity is around 500–1000% of the dry weight of the soil [1,8,14]. An average of around 2772 t C/ha is stored in undisturbed peat forest, with a further 186–306 t C/ha in biomass [15]. Noor [16] estimated the C stored in Indonesian peat systems to range from 300 to >6000 t C/ha, with carbon accumulation rates averaging 0.74 t/ha/y [8]. Worldwide, one-third of the carbon stored in peatlands is in the form of soil carbon, most of which is buried in the deeper catotelm zone (the saturated anoxic zone [17]). If managed incorrectly, the tropical peat ecosystem has the capacity to significantly accelerate climate change through release of globally significant carbon emissions into the atmosphere [18].

Peat ecosystems are fragile and prone to degradation because of their unique physical, chemical, and biological characteristics, which are very different from those of mineral soils [9,19]. Tropical peat soils are subject to irreversible drying, and have sensitivity to subsidence, low bearing capacity, low fertility and a limited number of microorganisms [9,12]. These specific characteristics are also influenced by a high diversity of spatial and vertical conditions which are determined by maturity level, thickness, the type of substratum, and the level of enrichment from the surrounding environment [12]. In Indonesia, wet tropical peatlands lie in lowland areas that usually have high rainfall, so under natural conditions they have poor drainage, leading to permanent inundation and anaerobic conditions with an acidic soil substrate [7,20,21].

Tropical peat ecosystems also provide livelihoods for local communities in agricultural cultivation and fishing, both of which are carried out in accordance with local wisdom [22]. Since ancient times, several local communities such as the Dayak and Banjar tribes in Kalimantan have practiced shifting cultivation, especially on shallow peat sites on river banks or coastal areas. These areas tend to be more fertile and also more accessible [13,22]. Peatlands have also been used for plantations since the 18th century when the colonial government increased exports of spices and plantation products. One of these examples is the existence of rubber plantations, since 1920, in the Anjir area on the border between Central Kalimantan and South Kalimantan [16].

Around 13 Mha of Indonesian peat swamp forest is estimated to be in a degraded condition [23,24], due to agricultural activities, logging and other land conversion activities [25]. These activities are generally preceded by the creation of canals [2,26] to access and regulate water systems so that the cultivated plants can grow well. However, the construction of canals causes disturbance to the natural hydrology of the peat [26], draining the upper layers, increasing surface water runoff and reducing water-storage capacity. These canals cause the ground water level to drop substantially in the dry season, which results in peat oxidation, accelerates peat degradation, increases carbon emissions and increases fire risk [25,27,28]. These impacts are contributing to the rapid global climate change [27].

The installation of canals causes changes in the physical properties of peat such as increasing its surface bulk density [29] and soil respiration (oxidation) rate [29], and encourages irreversible drying. These changes in soil physical properties have been reported to reduce the success of revegetation of degraded peat areas [30].

Another key aspect of peatland degradation is its increased susceptibility to fire. Dohong, et al. [31] found that peatland drainage directly increases its vulnerability to repeated fires. Fires are the source of enormous $CO_2$ emissions. For example, peatland fires in Central Kalimantan in 1997 were estimated to release 0.19–0.23 Gt carbon emissions over a burned area of 0.73 Mha [32]. The study of Wijedasa [33] revealed that repeated fires would result in increased carbon emissions, which was also directly correlated with the increase in bulk density (BD) in burned areas.

Repeated fires also result in reduced biodiversity [33] and slow the speed of peatland recovery, with a direct relationship between frequency of burning and recovery time [34]. This speed of recovery is influenced by several factors, including deterioration in soil physical properties [29,33], damage of seed sources and of parent trees [34,35] and decreased natural species diversity/density of the remaining stands [36,37]. Changes in the microtopography of peatlands also exacerbate the damage after burning, with Lampela, et al. [38] finding that fires caused changes in peatlands' microtopography, such as the loss of the hummock layer to become hollows, which reduces drainage and causes natural species to be stunted.

Indonesia has a target to restore 2.67 Mha of degraded peat over seven main provinces, which experience fires every year [39]. Restoration consists of three main activities: rewetting, revegetation and revitalisation of livelihoods [39]. The main purpose of rewetting is to bring the water table back to near the soil surface by blocking both primary and tertiary canals. There are substantial costs involved because there are so many canals to be closed. For example, in Central Kalimantan, a Mega Rice Project (MRP) resulted in more than 4400 km of canals [26]. Rewetting infrastructure must be planned carefully including the location of the installation, the estimated impact, and the materials to be used, which must be of high durability and low cost [25,26].

Revegetation of peatland after burning is expensive at around 1100 USD/ha [40], and still does not guarantee the success of revegetation due to declining species diversity [41]. Repeated fires change the nature of peat, such that it supports good growth of only a few of the original range of species, including *Shorea balangeran* and *Dyera polyphylla* [30,40,42]. Revegetation activities are prioritised where fires have been repeated

more than twice because natural regeneration still occurs in most areas that have been burned only once [34,36].

Livelihood revitalisation is an important aspect of peat restoration. Communities around peatlands often have traditional wisdom gained from managing peatlands for more than 100 years [43]. The use of land management without burning is an option for minimising the use of fires. However, nonburning agriculture is difficult for farmers to adopt because it is much more capital and labour intensive, and often out of reach of small farmers [44]. The challenges of this revitalisation effort are very large, and include choosing the right type of livelihood according to the site, the need for innovations including financial and technological support, and the creation of market infrastructure [45].

Restoration attempts to improve the condition of the disturbed ecosystem until it recovers its former function [46], but for the peat swamp forest ecosystem, restoration is complex and dynamic [47]. It requires a period of time measured in decades, in addition to significant political will and support from all stakeholders. In this context, the people who live and carry out activities on peatlands have a central role in their restoration [2]. Restoration should be viewed as an investment in ecosystems, including its people [48], and needs to have clear and measurable goals that can be monitored at every stage [49,50]. The success of restoration is critical because peatland restoration is key to reducing fires and $CO_2$ emissions and thus mitigating climate change [51].

The goal of ecosystem restoration includes achieving the same plant species and diversity as found in the remaining forest ecosystems or similar reference sites [46], but defining reference sites is not easy because of the fast and dynamic changes that occur [50]. It is necessary to understand the ecological history of the ecosystem including the climate, topography, soil science, hydrology and biota of the system, in addition to having an understanding of the main species of peat swamp forests, and how they can deal with changing environmental conditions [47].

Although increasing attention is being paid to the restoration of peat swamp forest ecosystems and the science of ecosystem restoration has developed, the methods for implementing restoration of tropical peatlands have not received much attention. Understanding of the ecology of peat swamp forests is still at an early stage, and there is no balance yet found between the ecological principles that must be applied and how this translates to on the ground activities [52]. Peat swamp forests have a slow natural regeneration ability [34,45,53], requiring decades of effort to achieve the desired result. Another challenge is the extremely degraded condition of many peatlands, which requires specific actions to overcome these factors [2,52], and varies with peat type and thickness. Restoration practices on thick peatlands will not work on thin peatlands or vice versa, noting that restoration practices of thick peat is of greatest concern [54].

The key to peat ecosystem restoration is rewetting of the profile and revegetation of the forest as a source of peat-forming organic matter [55]. Hydrological restoration will result in a safe site for plant recolonisation [56], prevent peat soil oxidation and reduce carbon emissions [57], minimising further subsidence [27] and preventing land fires [58]. The practice of revegetation is not limited to planting activities but includes encouraging natural revegetation processes [59]. In addition to the technical aspects of hydrology and revegetation, socioeconomic aspects are also critically important for restoration.

In this paper, we evaluate the efforts of the Indonesian government to restore its peatland ecosystems. Several aspects are reviewed, including regulations, institutions, peat restoration planning, criteria and indicators, and the issues and successes that have arisen during the implementation of peat restoration in the field. We explore the factors influencing the implementation of peat restoration in Indonesia and how it can be improved in the future. We review the outcomes of peat ecosystem and restoration research, especially on the islands of Kalimantan and Sumatra. The authors consist of researchers, practitioners and academics who have published research and have practical experience in the field related to peat restoration in Indonesia.

## 2. Regulations, Institutions, and Planning in Degraded Peatland Restoration

*2.1. Regulations of Peatland Restoration*

There are various regulations relating to the protection and management of peat areas at different levels of the legal hierarchy, which have existed since 1960. Although not all of them are directly related, they still have indirect implications for the peat ecosystem and its management. The hierarchy of the legal system of peatland restoration in Indonesia is presented in Figure 2. Of the national laws, Law Number 5/1960 [60] concerning Basic Agrarian Law sets out land tenure rights, including the rights of indigenous peoples in peat areas. Furthermore, Law Number 5/1990 [61] sets out the conservation of biological natural resources and their ecosystems and was relevant to peatland that is located in conservation areas. Law Number 41/1999 [62] sets out the management of peat in forest areas, and Law Number 18/2004 [63] sets out a reference for plantation commodities on peatlands. Law Number 26/2007 [64] concerns spatial planning and its implications for the hydrological unity of peat and its suitability. The next law that is most specifically related to the management of peat ecosystems is Law Number 32/2009 [65] on Environmental Protection and Management.

At the government regulation (PP) level, the government regulation that explicitly provides regulations related to peat ecosystems is PP 71/2014 [66] on the protection and management of peat ecosystems mandated by Law Number 32/1990 [67]. At the next legal hierarchy level, the initial arrangement on peatland is Presidential Decree (KEPPRES) Number 32/1990, which regulates the protection of peat areas, specifically those with a peat depth of three metres or more in the upper reaches of rivers and swamps that need to be protected. KEPPRES Number 82/1995 [68] established the Mega Rice Project (MRP) in Central Kalimantan, whose ambition was to convert one million hectares of peat and swampland for rice cultivation [69]. This was later realised to be a mistake and ended in total failure. The development of one million hectares of peatland areas became a dark history of peat management in Indonesia, as it focused on economic development only, without considering the value of sustainability.

More than ten years later, the government began to focus on peat rehabilitation, which was initiated through Presidential Instruction Number 2/2007 [70] concerning the Rehabilitation and Revitalization of Peatland Development Areas in Central Kalimantan and then followed by various other policies that emphasised the sustainability of the peat ecosystem, one of which was PP 71/2014 on the protection and management of peat ecosystems. PP 71/2014 specifies the peat depth and peat hydrological unit regulations in detail.

At the ministerial level, the restoration of peat ecosystems is regulated by the Ministry of Environment and Forestry Regulation Number 16/2017 [71] which set out the technical guidelines for restoration of peatland ecosystem function. It is intended to provide technical guidance for the national government, regional/provincial governments, communities (including indigenous people), and those responsible for businesses and or activities in restoring the function of peat ecosystems. In 2019, MoEF issued Ministerial Regulation No. 10 of 2019 [72], setting out the management of the peat dome peak based on the peat hydrological unit. This regulation serves as technical guidance for integrating efforts to conserve and manage the function of damaged peatland ecosystems and maintain their hydrological function.

Various regulations that directly or indirectly regulate peatlands and their hierarchy, from the Basic Constitution to the detailed rules set forth in ministerial regulations, are presented in Figure 2.

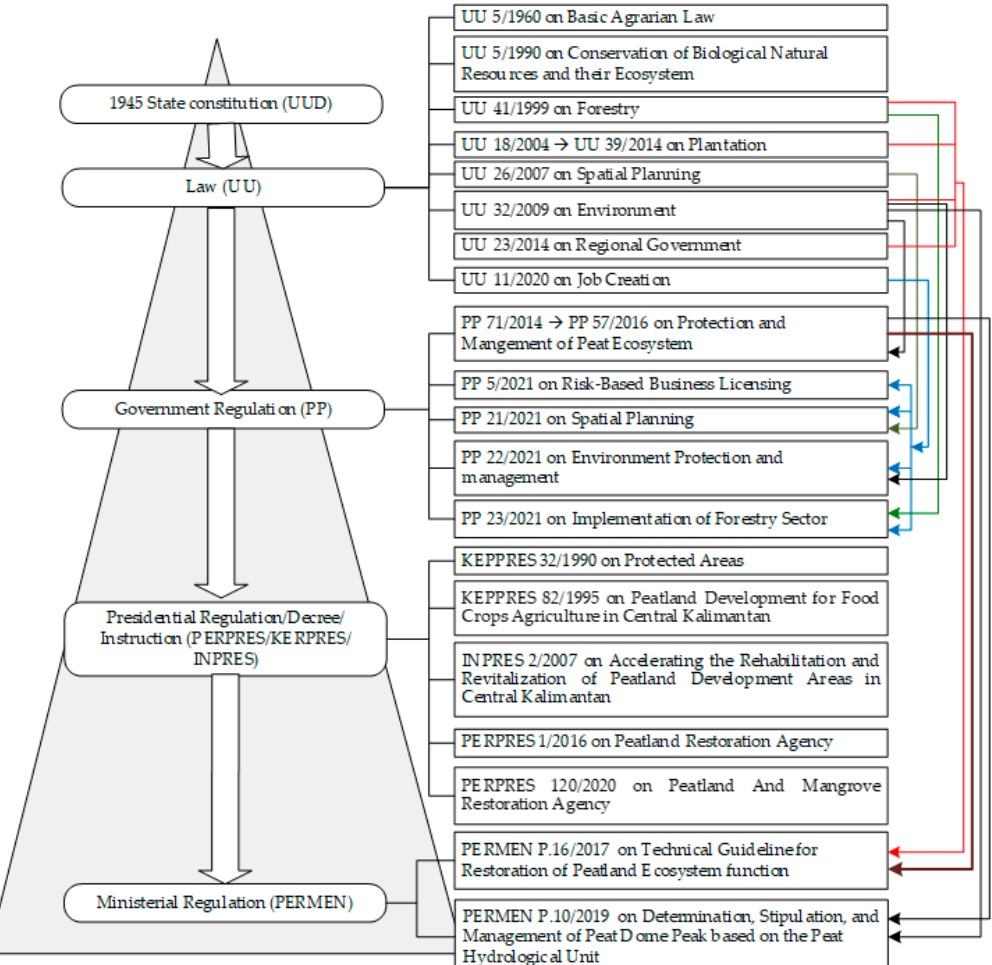

**Figure 2.** Hierarchy of the legal system of peatland restoration in Indonesia. The legal system includes 8 laws, 5 government regulations, 2 presidential decrees and one presidential instruction as the legal basis for 6 decades of peat management and restoration. (The colour of the arrows shows the derivative rules).

The milestone of peatland restoration regulation is the enactment of Presidential Regulation Number 1 of 2016 [73], setting out the establishment of *Badan Restorasi Gambut* or the Peatland Restoration Agency (PRA) to accelerate peat restoration due to the forest and land fires of 2015. In 2020, its role was expanded to include mangrove restoration through Presidential Regulation number 120 of 2020 [74]. One of the functions of this agency is to facilitate the implementation of peat restoration and efforts to improve community welfare in peat restoration work areas in the Provinces of Sumatra, Kalimantan and Papua.

The latest regulation is Law Number 11/2020 on Job Creation, which revises Law Number 41/1999 on Forestry and Law Number 32/2009 on Environmental Protection and Management. The issuance of this law was followed up with the issuance of several government regulations. Indeed, many people think that the issuance of this law and its derivatives, whose main spirit is to open up investment and create jobs by simplifying previous regulations, will threaten the sustainability of the peat ecosystem. Elimination of strict liability for environmental destruction and changes in environmental impact analysis procedures as regulated in Law Number 32/2009 concerning Environmental Protection and Management, in addition to eliminating quantitative limits on forest adequacy in a watershed at a minimum of 30%, including changes to the licensing system that simplifies business licensing according to the rules in Law 41/1999 on forestry, are considered to ignore the principles of caution. This will threaten the sustainability of peatlands and return them to the unsustainable peatland management practices of two to three decades

ago. There are many suspicions that ecological interests have been defeated by economic interests, which will later potentially cause damage to watersheds because restrictions are only based on debatable normative conditions.

From the legal aspect, it appears that support for peatland restoration efforts is very strong, with a tiered legal basis from the state constitution to regulations of detailed technical instructions (as presented in Figure 2). The existence of eight laws, five government regulations, two presidential decrees and one presidential instruction that have been the legal basis for more than six decades shows that peat management and restoration is a strategic issue. However, when viewed from the dynamic direction of the regulation outcomes, it appears that peatland is also the target of the political interests of each ruling government regime.

Most importantly, regardless of the government regime, the application of the legal system should not only be seen as a procedural arrangement, but also as a substantial arrangement concerning long-term environmental interests. A narrow interpretation of the legal system can harm the environment and ultimately eliminate people's rights to a good and healthy environment, as guaranteed by the 1945 Constitution, due to abuse of the law itself.

### 2.2. Institutions Involved in Peatland Restoration

Successful restoration requires good governance that encourages and synergises the roles of multiple stakeholders to be more effective in achieving the restoration goals [75]. Many parties play a role in peat restoration; nonetheless, some governance issues still exist, including disintegrated restoration planning and implementation, a lack of coordination between stakeholders and a complicated bureaucracy. These issues contributed to the underachievement of the restoration project, including its ability to control fires [76].

The institutions involved in peat restoration in Indonesia can be depicted in a stakeholders' power–interest matrix, as shown in Figure 3.

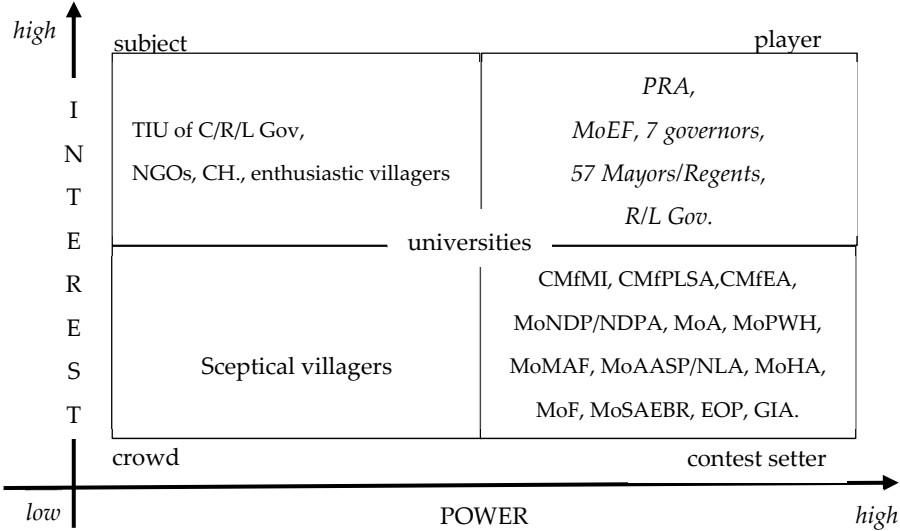

**Figure 3.** Power–interest matrix of stakeholder analysis of the acceleration of peat restoration in Indonesia. Description: PRA (The Peatlandt Restoration Agency); MoEF (Ministry of Environment and Forestry); R/L Gov (Regional/Local Government); CMfMI (Coordinating Ministry for Maritime and Investment Affairs); CMfPLSA (Coordinating Ministry for Politics Law and Security Affairs); CMfEA (Coordinating Ministry for Economy Affairs); MoNDP/NDPA (Ministry of National Development Planning/National Development Planning Agency); MoA (Ministry of Agriculture); MoPWH (Ministry of Public Works and Public Housing); MoMAF (Ministry of Marine Affairs and Fisheries);

MoAASP/NLA (Ministry of Agrarian Affairs and Spatial Planning/National Land Agency); MoHA (Ministry of Home Affairs); MoF (Ministry of Finance); (MoSAEBR (Ministry of State Apparatus Empowerment and Bureaucratic Reforms); EOP (The Executive Office of President); GIA (Geospatial Information Agency); TIU of C/R/L Gov (Technical implementing unit of central/regional/local government); NGOs (Nongovernmental Organisations); CH (concession holders).

Figure 3 shows that most of the stakeholders identified for peat management in the power–interest stakeholder matrix are those with high power. However, the key players only include the PRA, MoEF and the regional governments of seven provinces. Although numbering only a few, in terms of authority with the support of four laws and one presidential regulation, these stakeholders are very powerful and can determine the success of peat restoration.

Furthermore, villagers are divided into two groups. Sceptics are found among the wider public, with low power and interest, whereas villagers who are enthusiastic about the peat restoration programme are categorized as subjects with low power but high interest. These enthusiastic villagers have low capacity to achieve goals, but can become influential by forming alliances with other stakeholders. These stakeholders can often be very helpful so relationships with these stakeholders must be maintained properly.

The governance of peatland management and restoration has proven to be a challenge that focuses on the bureaucratic structure and patronage politics that are not yet optimal [77]. Consistent political will and effective communication are needed to ensure optimal peatland restoration. All stakeholders must communicate effectively so that the restoration plan can be collectively accepted and implemented. Therefore, well-organised institutions through government initiatives are needed.

Since peat restoration was made one of the national priority agendas in 2016 [78], the PRA was formed [79] with a target to restore burned peatland areas covering approximately two million hectares across seven provinces, namely Riau, Jambi, South Sumatra, West Kalimantan, Central Kalimantan, South Kalimantan and Papua. Through this initiative, the Government of Indonesia gave a strong commitment to peatland restoration by supporting and coordinating the key institutions and implementing teams at the national level (including nearly 14 ministries and central government agencies), in addition to regional and local levels [80].

The PRA is critical for strengthening the effectiveness of peatland restoration. The PRA is led by a head who is directly responsible to the President. The PRA is assisted by the secretary of the agency, several technical deputies, a group of experts and many staff. The PRA also is supported by several ministries and central government agencies through a technical steering committee. In addition, the PRA is also given the authority to independently manage its budget [80]. Furthermore, in coordinating and facilitating activities at the provincial level, the PRA is assisted by the Regional Peat Restoration Team that reports to the Governor. The Regional Peat Restoration Team coordinates and facilitates field activities from the district to village levels [79].

The PRA cooperates with all parties at the national and regional levels to realise the acceleration of peat restoration actions (see Figure 2). The stakeholders involved are technical units from vertical government institutions, regional government agencies, local governments, communities, private parties, universities and NGOs [79]. These stakeholders carry out peat restoration work as actors according to their duties, functions and responsibilities. In concession areas, in production forest areas, restoration work is carried out by concession holders. In conservation forest areas, restoration work is carried out by the area conservation management technical unit of MoEF. By comparison, restoration work for non-forest areas, non-concession production forest areas and protected forests is carried out by local governments through the Co-administration Task Force scheme [80].

The position of PRA as a central institution to coordinate and harmonise with central and local government institutions has been strengthened through the issuance of sev-

eral regulations by the relevant ministries. For example, the Minister of Environment and Forestry in 2020 issued regulation P.8 [81], which contains the assignment of some government affairs in the environment and forestry sector for peat restoration activities to seven governors. The mandate in the ministerial regulation clarifies the position and responsibility of each provincial government in assisting the PRA.

## 2.3. Peatland Ecosystem Protection and Management Plans

Peatland Ecosystem Protection and Management Plans (RPPEG) are prepared at the national, provincial and district levels by the central government Ministry of Environment and Forestry, provincial Governor and the district Regent, respectively. The RPPEGs are prepared hierarchically, with the lower-level plan referring to the higher-level planning document. RPPEGs' contents address peat ecosystem exploitation, control and maintenance, in addition to climate change adaptation and mitigation.

An inventory of the peat ecosystem is the first step during the planning stage, followed by the classification of the peat ecosystem's function into two categories: protecting the peat ecosystem and cultivating the peat ecosystem. A map of the function of the peat ecosystem and its current state of use shows the consequences and challenges that can affect peat ecosystem utilisation by various sectors, regions and communities.

In this regard, the plan for protecting and managing peat ecosystems considers the diversity of ecological characters and functions, population distribution, natural resource potential, local wisdom, community aspirations, climate change and spatial planning to preserve peat ecosystems.

The compilation of guidelines for the formulation, determination and amendment of the RPPEG is mandated by Article 19 of the Government Regulation Number 71/2014 [66] concerning the Protection and Management of Peat Ecosystems. These guidelines are intended to guide the Government and Regional Governments in creating, determining, amending, monitoring, and assessing the RPPEG and finance. The Directorate of Peat Damage Control, under the direction of the Directorate General of Pollution Control and Environmental Damage, is currently drafting a new regulation for the Minister of Environment and Forestry on preparing, determining and amending the RPPEG.

The Directorate of Peat Damage Control is preparing the National RPPEG after the MoEF issued Decree Number 130/2017 [82], which mandated the creation of the National Peat Ecosystem Function Map on a scale of 1:250,000. This decree is also the mechanism for implementing the MoEF's guidelines for the preparation, determination and amendment of the RPPEG. Data and information collection, and analysis, in addition to the preparation of potentials, problems, policy directions, strategies, programmes and activities related to the utilisation, control, maintenance, and adaptation and mitigation of climate change from the peat ecosystem are all part of the RPPEG preparation.

The ministry has also held public consultations on the drafting of the National RPPEG to increase the understanding of stakeholders regarding the preparation of the RPPEG, particularly for the government and local governments. The National Planning Agency (Bappenas), the Ministry of Agrarian Affairs and Spatial Planning/National Land Agency (ATR/BPN), the Ministry of Public Works and Housing (PUPR) and the Ministry of Home Affairs are the primary players participating in the central government's perspective. In addition, the aspects of local government include environmental authorities in provinces and districts/cities with peat ecosystems.

The Directorate of Peat Damage Control established the RPPEG KHG model to create a pilot project for planning protection and management at the level of the peat hydrological unit (KHG). There may be multiple KHG in a single district, each with varied conditions, resource potentials and ecosystem concerns, necessitating various measures to maintain and manage peat ecosystems. As a result, the RPPEG at the KHG or RPPEG KHG level is critical and must be considered when preparing the District RPPEG. In 2018, the Kapuas River–Terentang River RPPEG KHG model was created in Kubu Raya Regency, West Kalimantan Province.

Based on the above findings, rather than just setting procedures, the application of the legal system needs to prioritize substantial arrangements regarding long-term environmental interests. A restrictive interpretation can affect the environment and, ultimately, deprive people of their rights to a safe and healthy environment. Only the PRA, the MoEF and the regional governments of seven provinces are major stakeholders. These stakeholders are extremely influential and can determine whether or not peat restoration will be successful. However, the involvement of all stakeholders, especially the on-site villagers, is also a strategic determining factor.

## 3. Implementation of Peatland Restoration

### 3.1. Criteria and Indicators for Peatland Restoration

Methods for monitoring peatland vary depending on the extent of the peatland, the nature of anthropogenic disturbances, planned restoration interventions, objectives, accessibility, available resources and environmental targets' parameters [83]. It has been suggested that monitoring should be conducted periodically and focus on at least two reference conditions and on at least two variables within each of three ecosystem attributes of: species diversity, vegetation cover, structure, and biomass, and ecological function, such as nutrient pools and cycling, soil organic matter and mycorrhizae [84]. Monitoring can be used to inform the improvement of restoration activities [85]. Peatland restoration covers many aspects and multiple stakeholders with a range of interests, which can cause tension when there are trade-offs that need to be made between economic, social and environmental outcomes. This can generate intense disagreement between stakeholders interests (company concessions, communities, local governments, etc.); therefore, the success of restoration will be determined by how these varied agendas are harmonised, in addition to better governance and technical capacity building [86].

Some authors have suggested that total phosphorus, number of bacteria and fungi, and peat depth can be used as indicators of peat restoration status [14]), whereas others have suggested that key considerations must include restoration of closed cycles of organic matter, carbon, water, nutrient energy production, and integrated crops [8]. A consolidated set of criteria and indicators for peatland restoration is presented in Table 1.

**Table 1.** Criteria and indicators for the success of tropical peatland restoration.

| | Criteria | | Indicator | Reference Level (Healthy Peatland) | Details and Source |
|---|---|---|---|---|---|
| 1. | Ecosystem function | 1.1 | Protected areas | • No canals, water close to eat surface, extensive closed canopy. | Site is degraded if the water table drops due to drainage, resulting in peat oxidation and exposing pyrite and quartz layers [66]. |
| | | 1.2 | Cultivation areas | • Implementing paludiculture systems<br>• Hydrological rehabilitation (canal blocking/rewetting)<br>• Water table less than 0.4 m below peat surface. | Paludiculture can be used to rehabilitate all areas of degraded peatlands, except for areas severely degraded and subject to regular or prolonged flooding [87]. |
| 2. | Peat soil improvement condition | 2.1 | Chemical properties | • pH ranges from 3 to 5 | Deforestation, drainage and exposing pyrite can increase acidity (to pH levels of 2–3). [88,89]. |
| | | | | • C concentration around 500–550 mg Cg$^{-1}$, C/N ratios around 30 | Degraded areas have a C concentration higher than the pristine area; the C/N ratio increased by around 20 units with degradation [90,91]. |
| | | | | • Typical values for N, P, K and CEC in non-degraded condition tend to be higher than degraded one. | Revegetation of post fire peatland improves chemical soil properties. Over time, this leads to an improvement in total N, P, and K, as well as CEC [92]. |

**Table 1.** *Cont.*

| | Criteria | | Indicator | Reference Level (Healthy Peatland) | Details and Source |
|---|---|---|---|---|---|
| | | 2.2 | Physical properties | • High water holding capacity: peat can hold up to 500–1000% of its weight of water.<br><br>• Bulk density of 0.02–0.21 g cm$^{-3}$<br><br>• Natural forest soil depth slowly increases (no subsidence) | Drainage and peat drying can irreversibly reduce its water holding capacity [93]. Water holding capacity of the intact area (302%) was higher than that of the degraded area (196%) [49]. Peatland degradation increases the bulk density and reduces the value of hydraulic conductivity [27,89]. The drainage of peatland leads to peat subsidence. Disturbance can result in subsidence rates of more than 35 cm in the first five years [94]. |
| | | 2.3 | Biological properties | • Intact: $4.3 \times 10^6$ cfu g$^{-1}$, Oil palm area $1.1 \times 10^6$ cfu g$^{-1}$, and restoration area $2.4 \times 10^6$ cfu g$^{-1}$<br><br>• Soil fungi abundance of $4$–$11 \times 10^5$<br><br>• Soil macrofauna abundance >353 ind/m$^2$ | The natural forest has a culturable bacterial population that is higher than that of the degraded area [49]. The concentrations of microbial C and N were highest in the swamp forest and decreased following degradation level [90]. Natural peat forests hosted more diverse microbes than disturbed peat forests. Saprotrophic fungi were in greater abundance in natural peat forest, whereas phototrophs fungi tended to be more abundant in disturbed peat soil [95]. Revegetation activity affects the abundance of fungi and macrofauna in post-fire peatland. The abundance of soil fungi increases with understory cover) [92]. |
| 3. | Hydrological condition | 3.1 | Water table fluctuation | • Water table in natural forest remains at less than 0.4 m below soil surface | The average depth of the water table for intact land (Sebangau National Park) is around 0.4 m during the normal rainy season [26]. The water table fluctuation is lowest in the natural forest and increases following degradation, but the fluctuation depends on the rainfall [26,49]. |
| | | 3.2 | Hydraulic conductivity | • Hydraulic conductivity of tropical peatland is typically more than 10 m d$^{-1}$ | Hydraulic conductivity declined by 67% with conversion to agriculture [27,96]. |
| | | 3.3 | Potential redox (Eh) | • Potential redox <200 mV | The Eh value is strongly influenced by the water table depth at the measurement point [97]. |
| | | 3.4 | Evapotranspiration (ET) value | • ET of natural peat swamp forest tends to be close to potential evaporation. | ET values were around 3 mm/day in Sarawak [98]. As the vegetation becomes less dense and the surface dries out through degradation, ET declines. |
| 4. | Diversity of flora | 4.1 | Diversity of vegetation | • High diversity of vegetation species is characteristic of natural forest—a flora survey in Sebangau National Park found 103 taxa in 51 families [99]. | A survey of the nonforested area of the ex-MRP found 79 species of woody plants [36]. Edge effects caused by peat forest fragmentation significantly reduced tree species diversity and species richness. A total of 24 species in an interior site were not found near the edge site. Peatland restoration should be conducted to reduce forest matrices and the edge effects [100]. |

**Table 1.** *Cont.*

| | Criteria | | Indicator | Reference Level (Healthy Peatland) | Details and Source |
|---|---|---|---|---|---|
| | | 4.2 | Spontaneity of regeneration | • Natural forest recovers quickly after disturbance | Vegetation survey at Block A North-West ex-MRP, 13 years after being abandoned, was colonized by 8805 individual woody plants, comprising 6085 saplings and shrubs and 2720 trees. The mean density of woody plants was 0.09 individuals/$m^2$. The species of *Combretocarpus rotundatus* was more abundant than other tree species, including (*Cratoxylum glaucum*, *Cratoxylum arborescens*) and shrub species (*Melastoma malabaricum*) [36]. |
| 5. | Diversity of fauna | 5.1 | Ant community | • 5–10 ant genera are typical in natural forest | In the HCV area of oil palm plantations in the peatlands of Central Kalimantan, 23 genera of ants were found. The ant community was dominated by *Anoplolepis gracilipes*, an invasive species common in oil palm plantations but usually absent from high quality undisturbed forest, and its presence in HCVs is therefore indicative of highly degraded habitat [101]. |
| | | 5.2 | Avifauna | • Intact logged forest had 69 avifauna species | Degraded forest fragments and nonforest regrowth had reduced species numbers, with 36 and 32 species, respectively [41]. |
| | | 5.3 | Mammals | • The number of mammals species recorded from the peat swamp forest of Southeast Asia is 123 species [37]. | Loss of habitat leads to commensurate loss of mammal species with significant positive relationships between mammal species and canopy cover, canopy height, number of plant species, number of large trees (>30 cm) and number of fallen large trees (>30 cm) [37]. |
| | | 5.4 | Herpetofauna | • Herpetofauna species (amphibian and reptiles) are associated species in wet ecosystem. | A survey of the secondary forest area of KHDTK Tumbang Nusa, found 9 species of herpetofauna [102]. |
| 6. | Carbon stock | 6.1 | Below-ground carbon stock and emission | • Natural forest supports carbon accumulation at low levels | Loss of peat soil carbon stock occurs due to accelerated oxidation caused by drought and fires. Degraded peat soil can experience subsidence of 5 cm $y^{-1}$, equating to an average 5 year carbon loss of 178 $t$ $CO_2$-eq ha$^{-1}$ y$^{-1}$ [27]. |
| | | 6.2 | Dissolve Organic Carbon (DOC) | • Natural forest had levels of around 1–2 mg DOC g$^{-1}$ | Disturbed peat swamp forest had 50% [103] –70% [90] lower dissolved organic carbon compared to the undisturbed forest. |
| | | 6.3 | Aboveground carbon stock, recovery, and emission | • Above-ground biomass growth of 8–10 t ha$^{-1}$ in trees >10 cm | Forest biomass growth at forest edge sites reduced by 32% and 23–25% for tree diameters of 10–20 cm and >20 cm, respectively, compared to interior sites [100]. |
| 7. | Community livelihood improvement | 7.1 | Livelihood options | • Peat swamp forests can provide various alternatives for community livelihoods and can be managed sustainably. | Degraded peat swamp forests support less diversity of community livelihoods and most of the cultivation options (e.g., oil palm) tend to cause further degradation unless they can be achieved with a raised water table. |
| | | 7.2 | Household income | • Household income is derived from the sustainable use of peat swamp forest ecosystems. | Degradation can lead to unsustainable cultivation practices that may improve short-term income but result in longer-term loss of the capacity of the peat to support household incomes. |

**Table 1.** *Cont.*

| Criteria | Indicator | | Reference Level (Healthy Peatland) | Details and Source |
|---|---|---|---|---|
| | 7.3 | Community participation | • There is increasing community participation in sustainable management of land resources in peat swamp forests. | Unsustainable peatland use may result in exploitative practices whereby individuals stand to gain the most but hasten the degradation of the community resource. |
| | 7.4 | Economic | • Peat communities are economically self-sufficient, government subsidies to households low or absent, and there is good support to the economic sector [93]. | Unsustainable peatland use will result in longer-term loss of productive function and increased risk of fire, with the community requiring greater outside support. |
| | 7.5 | Social | • There is low migration from settlement projects, good employment opportunities, and enterprises are arising from forest use alongside effective local institutions for co-management [93] . | Unsustainable use and degradation may cause loss of the community resource, and with it the future opportunities for employment and sustainable enterprises. |

Indonesia has an ambitious vision to restore 2 Mha of peatland. Restoration as a long-term effort needs targets and measures for the achievement of its success. Restoration success can be achieved when all aspects can be managed, including both environmental aspects (ecological ecosystem) and social aspects. Several aspects synthesised from the results of this study, including ecosystem function, peat soil improvement condition, hydrological condition, diversity of flora and fauna, carbon stocks and community livelihood improvement, are expected to support the implementation of restoration.

Some of the criteria included in Table 1 have been implemented in real terms as a tool for assessing the health of the restored peat ecosystem. The reference of ground water level (maximum 40 cm below the ground surface), the danger of revealing the pyrite layer and the extent of the protected area that must be maintained have been included in the Government Regulation no. 57 of 2016 concerning the protection and management of peat ecosystems, and Regulation of Pollution and Environmental Damage Control Directorate General no. 10 of 2018 on The Guideline for Assessment of the Success of Peatland Ecosystem Restoration. The regulation also accommodates the importance of improving the economy of the community around the peat ecosystem. Criteria for improving peat soil conditions have also been accommodated in Government Regulation no. 150 of 2000 concerning control of soil damage for biomass production.

### 3.2. Rewetting

The drainage of peatlands results in a lowering of the water table, followed by oxidation of the unsaturated peat, emitting $CO_2$ [104] and changing its structure and hydraulic conductivity. This results in subsidence and increases its susceptibility to fire in drought situations [105,106]. Canals also threaten forest sustainability because illegal loggers often use canals to access the forest and remove logs [26].

Restoring the function of peat ecosystems as net carbon sequesterers, biodiversity hotspots and high-capacity natural water storages requires effective water management [27,107]. The key to protecting and restoring peat is to ensure it is always fully wet [77,108,109]. Therefore, in restoring degraded peatlands, the first action taken is rewetting [110]. This effort is generally carried out through canal blocking and canal infilling. These techniques have previously been applied in temperate and boreal peat areas, whereas in the tropics it is mostly conducted by blocking of canals [25].

In Indonesia, canal blocking is typically undertaken through construction of a box-shaped dam [26], with front and rear structures made from log poles, and the centre section

filled with peat and compacted or filled with piles of sandbags. The middle of the dam is usually equipped with a spillway, making it easier for small boats to cross the canal [87]. The size of the dam depends on the width of the canals. Narrow canals tend to require small dams that are constructed from locally available materials, whereas large canals such as irrigation canals need a more advanced design and are usually constructed by government agencies [25].

In Kalimantan, rewetting of the ex-MRP started in 2005 by Wetlands International in Block A of the Mega Rice Project (MRP) area, by WWF-Indonesia in Sebangau National Park, and pilot rewetting in Block C of the MRP area by CIMTROP [26]. At that time, the MRP was terminated, and the peatland was degraded. In Sebangau National Park, rewetting was implemented to reduce drainage and keep the water table at a safe height to maintain the hydrological and ecological function [111]. The pilot canal blocking in Central Kalimantan demonstrated that the design of the box dam system needs to account for several factors, including elevation. Canals that are constructed perpendicular to the elevation contour require multiple dams to prevent overtopping. Canals that are constructed parallel to contour lines do not require as many dams. Dam construction needs to account for the difference between the height of water in the canal and the peat surface (Ritzema et al. 2014), with simulation modelling suggesting that canal blocking can be most effective when the vertical distance between dams is 0.6 m [112]. This study also revealed that the effectiveness of canal blocking at maintaining the groundwater level is directly related to the balance between evapotranspiration and recharge, and the length of the drought period.

In recent years, rewetting efforts through canal blocking have been intensified [113,114]. In tropical peatlands, rewetting was first carried out in 2013 through the Climate Change, Forests and Peatland in Indonesia (CCFPI) project in Central Kalimantan and South Sumatra (Ritzema, 2014). In these two provinces, rewetting was achieved by blocking canals that illegal loggers had used for transporting logs. In South Sumatra, the project built 12 small blocks on four canals commonly used by illegal loggers [115].

A rewetting trial was also carried out by constructing canal blocks in Tanjung Leban village, Bengkalis District, Riau Province, in 2014. Observations made until 2017 showed an increase in the groundwater level of about 60 cm at the observation point 100 m from the block. The water level stayed at no more than 80 cm below the peat surface in the dry season, compared to before the installation of the canal block, when it had reached 160 cm below the peat surface. Subsequently, on a broader scale, the JICA Partnership Project (JPP) promoted voluntary water management by local communities by constructing 12 canal blocks. These canal blocks raised the water level by one metre over an area of 2420 ha. The area was divided into five elevation zones, and a total of 12 canal blocks were built on the boundary of each zone using local wood and sandbags at a cost of 300 USD per block [116]. In Central Kalimantan, researchers experimented with using PVC pipe filled with peat soil to block the canals, using wooden frames as the supporting component. This type of canal block was expected to use resources more efficiently, and can be applied to a small-scale revegetation area [117].

In Palembang, South Sumatra, canal blocks were piloted in two projects with agrisilviculture and agrosilvofishery systems. Although only secondary channels were blocked, it regulated and helped stabilise water table fluctuations. During the dry season, the water table at the agrisilviculture location dropped to 60 cm below the peat surface, whereas at the agrosilvofishery site, it dropped to 40 cm. Although 40 cm below the surface is sufficient to meet government regulations, it still exposes 40 cm of dry peat (and up to 90 cm in an El Nino drought), which is susceptible to fires and oxidation [118], so a more sustainable maximum level would be 20 cm below the peat surface [119]. The effectiveness of peatland rewetting can be increased by blocking more canals around the targeted site [119]. Field experiments at Sungai Tohor Village, Kepulauan Meranti District, Riau Province, showed that a canal block can effectively increase the groundwater level by

70 cm within a radius of 170 m. The radius can fluctuate depending on hydrotopography conditions, and characteristics of peat, land cover and water discharge in the canal [120].

Rewetting efforts by plantation forest companies are carried out by constructing a canal block made of peat that is compacted using an excavator. This type of dam, having a width of 2–20 m, has been applied since 2015 in plantation forest concessions in South Sumatra and Riau and the buffer zone of Berbak National Park [87].

There has been no long-term research on the durability of different types of canal block, or their effectiveness for rewetting [87], but the recommended method for restoration is using locally available construction material. Construction is followed by nearby planting using local species that are adapted to inundated peatlands. The plant roots are expected to hold the peat soil and reduce erosion rates, and in the future will replace the function of the logs in the dam structure because logs will decompose over time [26,115].

Although rewetting efforts cannot instantly improve the hydrological characteristics of peatlands, they are anticipated to lead to a rise in the water table. Peat that is below the level of the water table will have a much lower decomposition rate and, in fact, will contribute to peat formation, in addition to reducing fire susceptibility [2,121,122]. Even after being drained for 40 years, changes in the hydrological function of peat are not entirely irreversible, and rewetting can occur [107]. To date, there have been few studies into the effectiveness of efforts to restore hydrological function on peatlands in Indonesia, but a key indicator that has been widely reported is the response of the water table to rainfall input [123]. A more detailed study after 23 years of rewetting found physical improvements in peat soil, such as increasing water storage capacity and hydrological buffering function [123].

Based on the above findings, successful rewetting on a wide scale must pay attention to the selection of canal blocking techniques and materials that are suitable for field characteristics. Utilisation of local materials from around the site will save costs while taking into account the sustainability of the peat ecosystem. Community participation and local government support are also needed due to essential functions of the canal for access and transportation related to community livelihoods. People who understand the goal of restoration will more readily facilitate rewetting efforts [45]. Implementers of rewetting activities must also work with the community to offer alternative livelihood options to acknowledge the fact that existing livelihoods will no longer be feasible in rewetted peat [124].

### 3.3. Revegetation

#### 3.3.1. The Types of Peatland Degradation Sites

Peat swamp forest is subject to several types of degradation, including areas that have been logged (private or company areas), and are now only vegetated with shrubs or ferns. These areas may also not have been burned, burned once or burned several times. Different areas are subject to different levels of degradation [125,126]. The increasing disturbance to the peat swamp forest ecosystem and the magnitude of the impact on the environment has prompted many studies to address this issue, especially in Central Kalimantan. In the period before the implementation of the MRP, research was focussed on (1) understanding the biophysical conditions and diversity, in addition to the process of formation of peat swamp forest [127–129]; (2) the impact of forest exploitation on vegetation conditions and several planting efforts in the framework for species enrichment in logged-over forests [130]; and (3) the potential for carbon storage in tropical peat ecosystems, which at that time had not received much attention [131]. The research focus changed after the MRP development was stopped in 1999, and the haze disaster that started in 1997 made Indonesia the second largest $CO_2$-emitting country after the United States [32]. Research institutions and universities, both from within the country and abroad, started conducting research in Central Kalimantan on a range of topics, including biochemical processes of peat soil, carbon sequestration and emission, hydrology and forest ecology [132–135] and various impacts due to MRP [26,136,137]. Research on peat swamp forest restoration at that

time was based on research on peat swamp forest ecosystems in sub-tropical areas [138], and the revegetation approach was mostly focussed on species testing, such as that conducted by the Banjarbaru Forestry Research Institute between 2000 and 2008, and by the University of Palangkaraya and others [139,140]. These revegetation efforts still have not resulted in a satisfactory outcome, unlike the situation in Sumatra and Malaysia [140–143].

The poor success of peatland revegetation showed that there are various factors affecting the revegetation effort. Before commencing revegetation, the existing conditions need to be identified and determined, including: (1) the condition of the peat soil [34,130,144]; (2) factors that inhibit natural regeneration [52] and regeneration potential from seed sources such as remaining stands, seed rain, underground seed stored and vegetative shoots [145,146]; (3) the physical, chemical and biological changes of the peat soil for supporting plant growth; (4) the characteristics of plant species which survived on degraded areas [147,148]; and (5) the autecology of peat swamp forest species [140,149,150].

The decision for revegetation action should be based on the damage characteristics, as presented in Table 2.

**Table 2.** Damage characteristics and choice of intervention for the success of degraded peatland restoration.

| No | Damage Characteristics | Intervention |
|----|------------------------|--------------|
| 1. | Logged over forest with complete structure | - Tending of natural regeneration |
| 2. | Incomplete structure of logged-over forests/bushes | - Enrichment planting |
| 3. | Peatland area that has been burnt once | - Tending of potential shoots and natural regeneration |
| 4. | Peatland area that has been subject to repeated burning | - Planting based on species at reference sites |
| 5. | Peatland with repeated burning and changing hydrological status due to drainage (the presence of canals) | - Planting to form a new ecosystem<br>- Planting to accelerate the formation of a "safe site" to provide a jump-start for succession. A safe site is a site condition which supports natural regeneration and provides sufficient seed source such as seed rain and seed bank. |

### 3.3.2. Revegetation with Natural Regeneration

Peat swamp forests can recover naturally after disturbance. The ability of forest regeneration is not only affected by the diversity, composition and structure of the remaining vegetation [126], but it is also determined by the ability of the remaining vegetation to produce propagules and the persistence of the soil seed bank. Soil seed banks are a biological legacy that function as an ecological memory of an ecosystem [151], in addition to impacting the regeneration potential [152]. Natural forest regeneration follows the process of colonisation, stand formation, growth, and survival. The colonisation process is influenced by light conditions, nearby vegetation, and the composition of the seed rain and soil seed bank [153].

Natural regeneration of peat swamp forests must be supported by various efforts to improve the ecosystem as a whole. One of the key aspects is to improve the hydrological condition through canal blocking [2,52]. This improvement in hydrological function is expected to trigger faster natural regeneration [105]. The potential for regeneration of peat swamp forest needs to be well understood so that we can determine or predict the progressive succession.

Rachmanadi, et al. [154] explored the potential for natural regeneration in peat swamp forests by looking at wildlings, seed rain and the soil seed bank at sites with different levels of ecosystem damage. They found that the species number, the number of individuals and

the diversity of wildlings tended to decline with distance from the forest and with increasing damage. Wildlings tended to be of pioneer species, except for *Diospyros bantamensis*, for which the fruit comprises part of the diet of the native orangutans [155], and thereby has a wide distribution. In addition, this species is known to be semi-tolerant to light and disturbance, and is commonly found in areas that have been logged [150].

Remnant forest is important for natural forest regeneration [156,157]. High biodiversity encourages the stability of soil conditions in the presence of various root architectures and types [158]. Low diversity at the forest edge indicates that there are barriers to colonisation of some of the species [159], and is typical of degraded peat, such as the ex-MRP area Block A [41], and in Sebangau national park [99].

Seed rain in peat forest areas with open canopy cover was reported to range from 1127.8 to 1155 seeds $m^{-2}y^{-1}$ [154], similar to the number reported by Blackham, Andri, Webb and Corlett [35]. Although the number of seeds is greater in degraded peat swamp forest than reported in many other systems, the diversity of seed types is low. Seeds that persist in the soil layer are mostly representative of the climax vegetation from the previous forest and which have been spread by birds and mammals. Seeds from pioneer species that dominate the forest edge tend to be small in size and cannot survive in the soil for long. The small size of the seeds can result in fragile natural seedlings which are vulnerable to environmental changes such as inundation [160]. Failure of natural regeneration can also be a result of low seed quality [161], high threat of predators, extended residence time, poor natural media for germination [162], and early succession competition from grass and ferns [36].

### 3.3.3. Revegetation with Assisted Regeneration/Planting

Vegetation that can best tolerate peatland conditions are generally those species that are native [163], but seeds from these species are poorly available [164] and also less attractive for cultivation because of the long time-to-harvest life and products that are less attractive than traditional agricultural or plantation crops [163]. Two species that are being grown locally are *Shorea balangeran* and *Dyera polyphylla* [165–167]. The seeds of both species can be harvested every year and have high germination success. Native tree species with abundant seeds include *Combretocarpus rotundatus*, *Melaleuca cajuputi*, *Tristaniopsis obovata*, *Alstonia pneumatophora*, *Ploiarium alternifolium* and *Calophylum hosei*. However, the germination rates of these species tends to be very low, and further studies on these species are needed to promote better germination success [168].

The peatland sustainability is largely determined by the engagement of, and sense of belonging of, forest communities [169], especially if they are empowered to participate in revegetation activities that will enhance their livelihoods. An example is the utilisation of *Eleocharis dulcis* (purun tikus) as a natural alternative for making plant pots to reduce the use of polybags or plastics (Figure 4). The communities are expected to play a role in the rehabilitation process. Because it is the source of their daily livelihood, if successful, they will be the main actors in helping to mitigate climate change, increase biodiversity and productivity, and mitigate forest fires [169].

According to Santosa [170], two factors are important to the success of assisted revegetation or planting on peatland: bulk density and inundation. Low peat bulk density can cause plants to collapse easily, and inundation can be detrimental to seedling establishment. Santosa [170] suggested that these issues could be overcome by adopting appropriate planting techniques such as chopping of the peat before planting to increase bulk density, and compacting the soil around the seedling after it is planted. Establishment of sunken beds on shallow peat and mounding on deep peat, and using seedlings with a minimum height of 50 cm, can help to mitigate losses due to inundation [165,170].

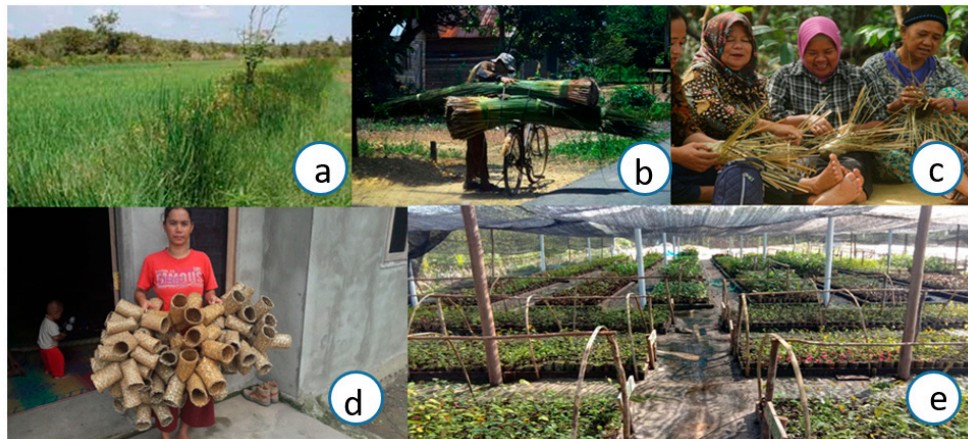

**Figure 4.** Utilisation of *Eleocharis dulcis* (purun tikus) as bio pots to reduce the use of polybags (Photo credit: Bastoni and Purwanto, 2021): (**a**) native *E. dulcis*; (**b**) transporting *E. dulcis* to the village; (**c**) organic pot production by women; (**d**) organic pots of purun; (**e**) utilisation of organic pots for native tree seedling production.

The growth of native tree species often stagnates in the first year after transplanting, as the roots are waterlogged in the rainy season and dry in the dry season, impeding access to nutrients and oxygen. One of the innovations that has been suggested is to improve oxygen flow to the roots using AeroHydro Culture technology [171], promoting aerial roots in the first 6–12 months. This may help native tree species to overcome their initial growth stagnation. Native tree species typically exhibit low growth rates when water levels are high for several months, and they run into oxygen deficiency, which suppresses their uptake of nutrients, mainly N and P. Mycorrhizal fungi are also important for stimulating roots under waterlogged conditions [172,173]. Preliminary outcomes from the AeroHydro Culture experiment in Riau and Central Kalimantan demonstrated good results in oil palm and *S. balangeran* plantations [171].

Many projects in the past 10 years have carried out revegetation activities but, to date, few results have been published on outcomes after planting [52]. Tending plays an important role in the plant's survival rate and growth [174], and needs to be carried out for at least two years on peatland. Common tending activities include replanting, weeding and protecting from pests, diseases and forest fires [174].

After planting, any mortality needs to be replaced through replanting, which has been conducted a maximum of three times in peatland projects: during the first 1–2 months after planting, at the end of the second year and at the beginning of the third year during the rainy season [175]. Graham, Giesen and Page [52] noted that weeds need to be controlled until the seedlings rise above the height of ferns (weeds) and sedges (about 1.5–2 m). New plants often die in the dry season due to drought and in the rainy season due to inundation, requiring replanting for up to three years after establishment. Plant mortality in the dry season mostly occurs on hummocks with a mortality rate of 6.13% per year, whereas in the rainy season, it tends to be lower-lying plantings, with mortality of 5.20% per year [176]. Plant mortality in the rainy season generally occurs in peat swamps that experience periodic inundation, where plant roots can be exposed to the air after inundation, causing post-anoxic injury [177].

Weeds are one of the main problems in peatland cultivation. The weed intensity on peatlands tends to be higher than that on mineral soils [178], with fast growth and production of 2–3 tons of dry matter/season/hectare [179]. Adequate weed control can minimise the competition with planted seedlings, preventing inhibition of the plant's growth [180]. In the early succession phase in natural forests, high light levels and the availability of water and nutrients can support opportunistic species to grow rapidly so that growth and survival of the target species can be reduced dramatically [181]. Weed control must also accommodate the light tolerance for each target species [52], with light-

demanding species (such as *C. rotundatus*, *A. pneumatophora*, *D. polyphylla*, *S. balangeran*, *S. leprosulla*, *C. galucum* and *T. glabra*) requiring more frequent weeding (every 2–3 months) to maintain a suitable light environment. For shade-tolerant species such as *Kompasia malaccensis*, weeding can be performed manually every six months at a radius of one metre around the plants. Weed control once every four months in the first year after *S. balangeran* planting can increase height growth by 40% and diameter growth by 50% compared to a non-weeded control [170]. Effective weed control also increased the average height of Sugi (*Cryptomeria japonica*) plants by 20% with each additional weeding that was imposed [180]. In Hinoki (*Chamaecyparis obtusa*), weeding restored plant growth in a short time due to delays in weeding in young plants [182].

Peatlands naturally have low fertility, and the addition of nutrients through chemical fertilisation will tend to increase peat decomposition [183]. Therefore, the best practice is to use organic fertiliser instead of mineral fertiliser, and only when plant growth is clearly limited by nutrients, or there is competition with weeds or other vegetation [175]. Organic materials such as wood ash can be applied at between 0.5 and 15 ton/ha [183]. Inoculation of local native forest plant seedlings with native mycorrhizal fungi can also help to improve the quality of seedlings so that they are better able to adapt and survive when planted out [168,184,185]. Mycorrhizal fungi improve the capacity of the plants to find sources of nutrients in peatlands, both in flooded and dry conditions, with good growth responses, both in the nursery and in the field. *Shorea balangeran* responds effectively to inoculation with ectomycorrhizal fungi in the nursery. Another of the native species, *Dyera polyphylla*, associates with arbuscular mycorrhizal (AM) fungi, having high AM colonisation and a high survival rate after one-year planting. Both tree species are widely planted (Table 3) and can adapt well to waterlogged conditions, as long as the seedlings are taller than the anticipated maximum water level (Figure 5.). Other promising native pioneer tree species include *Tristaniopsis obovata*, *Melaleuca cajuputi*, *Combretocarpus rotundatus*, *Ploiarium alternifolium*, *Alstonia pneumatophora*, *Metroxylon sago*, and *Tetramerista glabra* (Table 3).

Based on the above findings, the key to the success of peat revegetation lies in the (1) identification of the existing conditions, including the damage characteristics and the regeneration potential of sites; and (2) the silvicultural knowledge of native tree species while considering local wisdom, the active participation of local communities and the supporting technological inputs, such as application of non-burnt land preparation, AeroHydro Culture technology, mycorrhizal inoculation, paludiculture and utilisation of organic pots.

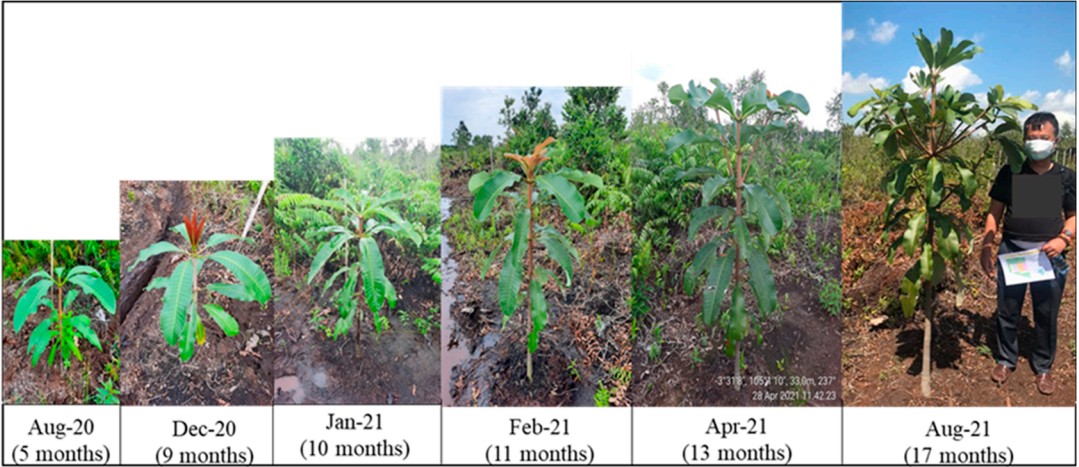

**Figure 5.** Growth performance of *Dyera polyphylla* in degraded peatland 17 months after transplanting in the field, South Sumatera, Indonesia. (Photo credit: Bastoni).

**Table 3.** Revegetation trial activities and development forest plantation in degraded ecosystem peat swamp forests in Indonesia.

| No. | Location and Area | Tree Species | Notes and References |
|---|---|---|---|
| 1. | Forest Research Institute (FRI) Palembang collaboration with International Tropical Timber Organization (ITTO), location plot at Sepucuk (20 Ha), South Sumatera | *Gonystylus bancanus*<br>*Dyera polyphylla*<br>*Tetramerista glabra*<br>*Shorea balangeran* | Initially, without rewetting at the time, drainage was constructed in plantations around the plot [87], |
| 2. | FRI Palembang collaboration with Peatland Restoration Agency (PRA), at Sepucuk (10 Ha), South Sumatera | *Shorea balangeran* | Agrosilvofishery, some part of the area was burned by fires. |
| 3. | Forest Management Unit (KPHP) V Pademaran Lempuing, FRI Palembang, Forest Research and Development Centre (FRDC-Bogor), in collaboration with The Mushroom Initiative (TMI), location at Pademaran, South Sumatera | *Dyera polyphylla*<br>*Shorea balangeran*<br>*Tristaniopsis obovata*<br>*Melaleuca cajuputi*<br>*Dryobalanops aromatica*<br>*Cratoxylum glacum*<br>*Fragraea fragrans* | Application of mycorrhizal fungi and use of organic pots. |
| 4. | FRI Banjarbaru, FRDC-Bogor, TMI, location at Tumbang Nusa (50 Ha), Central Kalimantan | *Dyera polyphylla*<br>*Shorea balangeran*<br>*Tristaniopsis obovata*<br>*Combretocarpus rotundatus*<br>*Melaleuca cajuputi*<br>*Alstonia pneumatophora*<br>*Vatica rassak*<br>*Calophyllum hose*<br>*Syzygium garcinifolia* | Application of mycorrhizal fungi and use of organic pots. |
| 5. | Katingan Peatland Restoration and Conservation Project, 1.23 Ha, Central Kalimantan | *Dyera polyphylla*<br>*Shorea balangeran*<br>*Melaleuca cajuputi*<br>*Alstonia scholaris* | Katingan is being implemented as an ecosystem restoration project by Rimba Makmur Utama, Ltd. [87]. |
| 6. | Padang Island (Meranti Archiphelago Regency) and Bengkalis Island, several 10,000 Ha, Riau Province | *Metroxylon sago* | Extensive sago plantation as a livelihood for forest communities [87]. |
| 7. | Tanjung Leban, Bengkalis Restoration Site, University of Riau, CIFOR, Global Landscape Forum, 5.25 Ha | *Dyera polyphylla*<br>*Shorea spp.*<br>*Hevea brassiliensis* | There is no hydrological rehabilitation, mixed species plantations [87]. |
| 8. | Dyera Hutan Lestari Ltd., 2000 Ha, Sei Aur, Jambi Province | *Dyera polyphylla*<br>*Alstonia scholaris* | Forest plantations, but abandoned after repeated fires [87]. |
| 9. | Tri Pupa Jaya Plantation (Asia Pulp and Paper-Sinar Mas Group), 2000 Ha, South Sumatera | *Dyera polyphylla*<br>*Alstonia scholaris*<br>*Palaquium burckii*<br>*Shorea leprosula* | Creation of a buffer zone between forest plantations and adjacent protected areas [87]. |
| 10. | Conoco Phillips Ltd., 200 Ha, Jambi Province | *Dyera polyphylla*<br>*Alstonia scholaris*<br>*Calophyllum sp.*<br>*Dryobalanops*<br>*Shorea pinanga* | Some tree species died after rewetting (Giesen and Sari, 2018). |
| 11. | Inhutani II Ltd. And Gajah Mada University, Segedong-Samandaka, 2220 Ha, West Kalimantan | *S. macrophylla*<br>*S. stenoptera*<br>*S. guiso*<br>*S. teysmanniana*<br>*S. compressa*<br>*S. balangeran*<br>*S. mangachapoi* | The location was not actively managed, and some natural regeneration of Dipterocarpaceae occurred [87]. |
| 12. | FRI Banjarbaru, FRDC-Bogor collaboration with AFOCO (Asian Forest Cooperation Organization), 4 Ha, Tumbang Nusa, Central Kalimantan | *Dyera polyphylla*<br>*Shorea balangeran*<br>*Combretocarpus rotundatus*<br>*Nothapoebe* cf. *umbelliflora* | *D. pollyphylla* and *S. balangeran* demonstrated good growth performance. |
| 13. | FRI Banjarbaru and Telkom Palangkaraya Ltd., 2 Ha, Tumbang Nusa, Central Kalimantan | *Shorea balangeran* | They followed the Re-Peat program (Purwanto, pers.comm). |

**Table 3.** *Cont.*

| No. | Location and Area | Tree Species | Notes and References |
| --- | --- | --- | --- |
| 14. | JSPS (Japan Society for the Promotion of Science) Hokkaido University and University of Palangkaraya, 2 Ha, Kalampangan, Central Kalimantan | *Shorea balangeran* | Seedlings were inoculated with ectomycorrhizal fungi [184]. |
| 15. | R&D Sinar Mas Forestry, Komatsu Ltd., and Forestry Research and Development Centre, Perawang, 16 Ha, Riau Province | *Shorea balangeran* *Melaleuca cajuputi* *Cratoxylum arborescens* *Camnosperma coriaceum* | *S. balangeran* grew well in peatland and is a potential candidate for pulp and paper [87]. |
| 16. | Tolan Tiga Indonesia Ltd., Barumun River, 10 Ha, Riau Province | *Shorea* spp. (Dipterocarpaceae) | High mortality [87]. |
| 17. | Great Forests Parks (Tahura) Orang Kayo Hitam, Berbak, 8000 Ha, Jambi | *Dyera polyhylla* *Metroxylon sago* *Tetramerista glabra* | Canal block had been completed [87]. |
| 18. | World Wide Fund for nature (WWF), Londerang site, 200 Ha, Jambi Province | *Dyera polyphylla,* *Artocarpus integra* *Mangifera indica* *Durio zibethinus* *Nephelium lapaceum* | This location was not adapted to full rewetting [87]. |

*3.4. Revitalisation of Local Livelihoods in Peatland Restoration*

3.4.1. Recent Livelihoods and an Overview of the Revitalisation Programme

The livelihoods of people on peatlands are often heavily based on natural resources and dominated by forestry, fisheries and agriculture [86]. Livelihood activities can have a major impact on land and forest cover changes [186], especially for agricultural development for industrial crops such as oil palm, acacia, and rubber that became the main drivers of peatland conversion [167]. It is also driven by the increasing population and the need for agricultural land to support their livelihoods.

People have had to adapt to changing environmental conditions in order to continue to generate their livelihoods. This includes moving to new locations and expanding the area that they work, working more intensively or increasing their number of livelihood options [187]. They can also adapt their livelihood options to improve their income. They usually employ up to three or more different options. Communities who initially made a living only by collecting natural resources began to switch and carry out cultivation activities. The types of plants vary greatly depending on the experience of the community and on their financial capacity (capital). Access to market and market demand is another important factor in choosing which crops to grow [188].

Plantation and agricultural cultivation on peatland require more effort, knowledge and capital than on mineral soils because only limited crops and livestock will survive in this environment. Basic knowledge of land management and the ability to choose suitable plant species are needed to be successful and profitable as a source of community income. Paludiculture has been promoted as a solution for utilising peat that has been rewet, and therefore to help improve the ecological condition of peatlands. Despite this, paludiculture options remain limited and the main developments on peatland continue to be oil palm and rubber [31]. Both of these crops cannot be grown with a high water table, so are not compatible with paludiculture [87,167]. Although both oil palm and rubber can survive in an area that is occasionally inundated, the peat needs to be drained for them to be productive.

In addition to oil palm and rubber, the community has developed several other crops. In Tanjung Jabung Timur Regency, Jambi, potential crops that have been identified include areca nut, coconut, liberica coffee, *Parkia Speciosa* (petai), *Shorea belangeran* and *Archidendron pauciflorum* (jengkol) [189]. Moreover, in Pelalawan district, Riau Province, pineapple, cassava, sago and coconut have been recommended because they are more environmentally friendly than oil palm and rubber [190]. The community has also recognised that

agroforestry systems with agricultural or plantation crops mixed with forestry trees may also improve outcomes. A demonstration plot introducing an agroforestry system model by planting mixed crops of fruits (*Nephelium lappaceum*, *Durio zibethinus* and *Mangifera indica*), plantation species (*Coffea liberica* and *Pinanga* sp), and forest trees (*Dyera costulata*, *Shorea balangeran* and *Illex cymose)* has been established in West Tanjung Jabung, Jambi Province [191]. Some models of agroforestry are considered quite profitable, both from an economic and ecological perspective [192]. They allow the community to generate cash income quickly, and have longer-term returns (within ten years or more) from annual sales of forest products. Sustainability, profitability, scalability of the market and acceptability to farmers were considered important in selecting plant species to be developed by the peatlands community in Central Kalimantan [188]. Based on these considerations, sago (*Metroxylon sago*), illipe nut/tengkawang (*Shorea spp*.), mangosteen (*Garcinia mangostana*), pineapple (*Ananas comosus*), banana (*Musa paradisiaca*), water spinach (*Ipomoea aquatica*), edible fern (*Stenochlaena palustris*), sweet melon (*Cucumis melo*) and dragon fruit (*Hylocereus undatus*) were recommended for cultivation in peatlands. Transforming traditional methods into a more sustainable system became one of the strategies of livelihood revitalisation for the community who lived near the peatland [119]. These can refer to zero burning peatland preparation, introducing the nondrainage peatland use management, and promoting eco-friendly and prospective commodities and environmental service towards sustainable peatland management. By 2020, more than 1000 community livelihood revitalisation packages had been launched by PRA and its partners, involving more than 29,600 community members in their programmes [193]. However, implementation at the site level needs to be optimised to improve livelihoods of people living around peatlands.

### 3.4.2. Challenges and Opportunities for Improving Livelihoods in Peatland Restoration

Peatland communities are the main target for peatland restoration activity. At the field level, the role of the local people is crucial to ensure continuity and successful restoration activity after the peatland restoration programme is finished. New livelihoods are needed to improve the community returns from the trade-off between peatland restoration and existing livelihoods [2,186,194,195]. Promoting suitable alternative livelihood options is an important component of peatland restoration [193]. However, there are a range of challenges and opportunities that need to be addressed to improve livelihoods through peatland restoration.

Changing existing livelihoods is challenging because communities are resistant to change and they prefer the status quo: illegal logging, building canals, and using fire for hunting and fishing [196]. Peatland canal blocks have been destroyed by people who were unaware or not accepting of rewetting activities [25,196]. Examining and embedding social dimensions in any restoration activity is challenging. Livelihood activities that cause the condition of peatlands to become increasingly degraded are still being performed by many people [31,196], including continued land conversion [188], which is contrary to the restoration programme. The local communities are conducting extractive livelihood activities that prioritise economic returns, because they need income to survive, and there is a lack of more sustainable livelihood alternatives [25,194]. Encouraging people to transition to new livelihoods is still difficult as there are few options that are competitive with existing livelihoods [197]. Additional funding is crucial to support the restoration programme [45] and to provide more prospective sustainable livelihood options in peatland.

While the existing, unsustainable, livelihood options continue to be developed, the resulting peatland degradation negatively impacts hydrological systems and vegetation recovery [193], leading to difficulties in revegetation and restoring sustainable livelihoods [45,110,196]. The degradation also continues to promote recurrent fires, putting the restoration programme at high risk of failure and loss of livelihoods [25,45,196].

Limited prospective commodities and markets from the paludiculture species are challenges in the development of improved livelihoods from the restoration programme. The concept and practice of paludiculture has already been implemented through various

programmes, but the scale-up and commercial purpose needs more effort. A vast range of timber and non-timber species have been recommended as paludiculture options to support the community's livelihoods for restoration on degraded peatland [58,87]. Unfortunately, only very few species provide significant economic returns, and the market remains limited [103,110,163,188,195]. Thus, existing crops that rely on drained peat still remain attractive compared to paludiculture species, and paludiculture species remain a low priority for the community.

Institutional and local capacity are also a challenge for revitalisation of livelihoods in peatland restoration. Complex decentralisation of government, land tenure, and property rights are barriers to ready adoption of new livelihoods [195]. Even though the PRA has put significant effort into revitalisation efforts, there are fewer support mechanisms for smallholder farmers in peatland [188]. Key issues that have been raised in previous studies include lack of community participation [45,103,195,197,198], lack of monitoring and evaluation [110], and the short duration of programmes compared to the long-term impact of restoration [45]. Another challenge is the lack of knowledge base of peatland restoration and lack of opportunity for sharing knowledge among stakeholders [195,197]. This also highlights the issue of communication and building networks in peatland restoration.

The PRA has established a significant number of revitalisation programmes, which can be explored to understand which programmes are successful and could be scaled up. The programmes that have not been successful can also be better understood so that they can contribute to improving future policy and programmes. There were 1214 revitalisation livelihoods programmes launched by the PRA up to 2020, including: (1) land-based programmes; (2) water and fishery-based resource programmes; and (3) environmental services programmes [193]. In addition to these PRA-supported programmes, paludiculture studies and initiatives from other stakeholders also provide a valuable resource for understanding what successes and failures have occurred through peatland restoration [110,163,188,192,196,199].

The development of mixed-farming (such as agroforestry, agrosilvofishery and agrosilvopasture) can potentially enrich the peatland restoration efforts and help the peatland communities to transition to fully restored peatland. Production of diversified commodities from mixed farming systems is a characteristic of traditional farming that has been practiced for generations [43,124] and has the potential to provide significant economic returns to growers [110,192]. In the context of food and livelihood security, smallholder farmers can implement mixed-farming systems in peatland to minimise risk, provide various sources of income and ensure food security throughout the year [188]. Adaptive agroforestry and adaptive paludiculture may help to reconcile improving livelihoods with peatland restoration.

Encouraging collaboration work and generating more opportunities for communities to participate in sustainable peatland management are the contextual options to ensure the revitalisation of livelihood programmes. Among stakeholders, the community around degraded peatland is one of the key successes for the long process of peatland restoration. Incentives for developing prospective livelihoods and commodities in peatland can stimulate communities to become actively involved in peatland restoration. Moreover, building a multi-stakeholder partnership in peatland restoration programmes is a potential means to improve the public's awareness and actions in exploring more sustainable livelihood options in peatland.

## 4. Conclusions

Although formal peat management in Indonesia started in the 1960s, it was inappropriate peat management, leading to degradation, which prompted the government to issue various regulations for fundamental improvements in peat governance. Peat governance has been improved in Indonesia through various regulatory instruments, including by following the 3Rs approach (rewetting, revegetation, and revitalization of local livelihoods), and institutional aspects of peat management, which are more oriented towards

sustainability than just focusing on economic considerations. Rewetting degraded peatland is predominantly achieved through canal blocking. Although canal blocking cannot instantly restore the hydrological function of peat, it can gradually lead to an increase in the water table. The effectiveness of canal blocks can be increased by paying attention to the construction and distance between canal blocking, and, most importantly, involving the community to prevent canal blocks being removed by peatland users unaware of restoration goals. The choice of intervention for revegetation action should be based on on-site damage characteristics. Assisted revegetation should consider the cost and threat of fire. Natural regeneration is still the main option for large-scale restoration at a reasonable cost. It is critical that community livelihoods are considered in the restoration effort, and it is imperative to ensure that communities have profitable livelihood options that are compatible with ecosystem restoration. At the local level, more comprehensive restoration activities that emphasise these livelihood benefits are important for encouraging community participation.

**Author Contributions:** T.W.Y.; D.R.; P.; M.T.; Y.I.; H.Y.S.H.N.; M.A.Q.; B.H.N.; B.W.; S.L.; P.B.S.; R.N.A.; E.S.; P.B.P.; R.S.W.; R.P.; W.H.; B.N.; B. and D.M. had an equal role as main contributors in discussing the conceptual ideas and the outline, providing critical feedback for each section and writing the manuscript. All authors have read and agreed to the published version of the manuscript.

**Funding:** This research was partly supported by ACIAR through grant number FST/2016/144.

**Institutional Review Board Statement:** Not applicable.

**Informed Consent Statement:** Not applicable.

**Data Availability Statement:** Not applicable.

**Acknowledgments:** We would like to thank ACIAR FST/2016/144 for providing financial support for this paper. We would also like to thank Andrea Rawluk from The University of Melbourne for useful comments and suggestions for the improvement of this manuscript.

**Conflicts of Interest:** The authors declare no conflict of interest.

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
