# Peer review of "Restoration of Degraded Tropical Peatland in Indonesia: A Review"

_land, doi:10.3390/land10111170_

Round 1

Reviewer 1 Report

The presented article is devoted to the actual topic of the restoration of drained bogs. The material is considered in sufficient detail and versatile from the standpoint of ecology, economics and sociology. The analysis carried out and the generalizations made are interesting and important for science, business and government bodies. The described experience is undoubtedly relevant outside the region to which the article is devoted.
The necessary corrections are presented in the attached file.

Author Response

Dear reviewer 1,

We thank you for your comprehensive comments and suggestions for our manuscript. Please find attached our response to the reviewer and the revised draft of the manuscript. 

Reviewer 2 Report

Manuscript Review: Restoration of Degraded Tropical Peatland in Indonesia: A Review

Summary

The submitted review summarises restoration activities, challenges, policies, and indicators for peatland ecosystems in Indonesia. The manuscript is an important contribution to our understanding of the efficacy of initiatives used to date, and considerations needed for future activities. The authors present opportunities to better integrate the ecological and social outcomes of peat restoration in Indonesia. The manuscript is thorough and very well written. I have made suggestions for minor edits/improvements below.

Suggestions for minor edits:

L30: ‘peatland’ to ‘peatlands’

L42: remove extra space between ‘Sulawesi’ and ‘(0.03)’

Fig. 1: add white/light background for legend and arrow/scale bar to make easier to read

L48-9: either ‘residue’ (singular) or ‘have’ (plural)

L57-59: is the surface layer more fresh than deep peat as it is less decomposed?

L80: replace ‘in’ with ‘under’ natural conditions

L79-81: consider moving this sentence to the previous paragraph, which focuses on environmental characteristics.

L91: Include the relevant region, e.g. ‘Around 13 Mha of Indonesian peat swamp…’

L120: consider rephrasing as ‘…to restore 2.67 million hectares of degraded peat over seven main provinces, that experience fires every year’

L137: change ‘peatland’ to ‘peatlands’ or ‘peatland ecosystems’, or similar (plural).

L144: consider rephrasing to make clearer, e.g. ‘Restoration attempts to improve the condition of the disturbed ecosystem until it recovers its former function’ or similar.

L147: consider rephrasing to make clearer, e.g. ‘…the people who live and carry out activities on peatlands have a central role in their restoration’

L159: replace ‘to restoration’ with ‘to the restoration’ and ‘increased’ with ‘increasing’ and ‘ecosystem’ with ‘ecosystems’

L177: replace ‘the peat ecosystem’ with ‘its peatland ecosystems’ or similar, to make clear you’re only referring to the peatlands managed by this govt.

L195 & 197 & 204: remove double space

Fig. 2: Include in the caption an explanation of the colours used, both in the boxes and arrows.

Fig. 3: The word ‘Interest’ on the Y-axis is cut off

L323: This sentence is a little confusing - perhaps ‘Most of the stakeholders identified for peat management in the power-interest stakeholder matrix are those with high power’

L425: include the word ‘and’ or ‘or’ before mycorrhizae

L418: ‘Indicator’ should be plural

L145-6: Consider rephrasing, e.g. ‘Monitoring can be used to inform the improvement of restoration activities’ or similar.

L430: change ‘but’ to ‘thus’ or ‘therefore, the success…’

L433: remove ‘indicators of’ (it’s repeated later in the sentence), replace with ‘some authors have suggested that total phosphorus…’

Table 1: This table is an excellent summary of data, however, the indicators are a little confusing to me – are they indicators that a peatland requires restoration activity (i.e. indicators of degradation) or indicators of acceptable peatland health? Or in some cases, just descriptions of current opportunities? I think it would be good to be both consistent and very specific here, using quantitative indicators wherever possible – as this table is a valuable part of the paper and I think will be of great interest to many readers.

A suggested format for this table, with some examples, is below (noting that where it isn’t appropriate to be specific about the reference level, you could state ‘site-specific’, and still include details from the literature in the final column). The ‘reference level’ column could also be rewritten as an indicator that restoration is required, e.g. in the first row, ‘No canals exposing…’ would be switched to ‘Canals exposing pyrite & quartz’ etc.

Criteria

Indicator

Reference level (healthy peatland)

Details & source

1.     Ecosystem function

1.1 Protective function

·     No canals exposing the pyrite & quartz layer

·     No reduction in area size or canopy cover

·     Water table <0.4m below peat surface during the rainy season

Site is degraded if the water table is more than 0.4 m below peat surface, exposing the pyrite and quartz layer [66].

2.     Peat-soil improvement condition

2.1 Chemical properties

pH range from X-Y

Deforestation and drainage can increase acidity (to pH levels of 2–3). When drained, aerial oxidation leads to the release of acidity and other plant toxins, especially aluminium, from peat. If the pH value is higher than 5.5, it will facilitate peat soil mineralisation [87].

C/N ratio in from X-Y

The degraded area has a C concentration higher than the pristine area; the C/N ratio increases following the degradation condition [88].

 2.2 Physical properties

Water holding capacity from X-Y

Peat can hold 500– 1000% of its weight of water. Drainage and peat drying can reduce its water holding capacity [90].

Water holding capacity of the intact area is higher than the degraded area, e.g. 302% compared with 196% [49].

Bulk density less than X g cm3

Peatland degradation increases the bulk density and reduces the value of hydraulic conductivity [27].

2.3 Biological properties

Bacterial population higher than X cfu g-1

(or ‘Bacterial population significantly more diverse than adjacent degraded area’)

The natural forest has a culturable bacterial population higher than the degraded area, 4.3x 106 cfu g-1 [49].

The concentration of microbial C and N were highest in the swamp forest and decreased following degradation level [88].

Soil fungi abundance of 4-11 x 105

Natural peat forests harboured more diverse microbes than disturbed peat forests. Saprotrophic fungi were in greater abundance in natural peat forest, while phototrophs fungi tended to be more abundant in disturbed peat soil [91].

Revegetation activity affects the abundance of fungi and macrofauna in postfire peatland. The abundance of soil fungi is 4 x 105 to 11 x 105, while the abundance of soil macrofauna is 353– 1038 id/m2. The abundance of soil fungi increases with understory cover (revegetation time) [89]

Soil macrofauna abundance >350 id/m2

L452: consider ‘…and high-capacity natural water storages’ to keep grammar consistent in this list.

L482: ‘peatland’ should be plural

Section 3.2: keep units of water depth consistent (either m or cm)

L530-531: change ‘effects’ to ‘effectiveness’

L542: replace ‘rewet’ with ‘rewetted’

Table 2: Include a definition of “safe site” either as a table footnote or in the main text.

L593: add ‘the’ or ‘its’ after “One of”

L612-613: Identify what each of these numbers refers to – e.g. are they reported for different years or different areas?

 L636: Change “the sustainability of peatland” to “peatland sustainability” or change peatland to peatlands.

Fig. 5 caption: move ‘after’ to read “17 months after transplanting in the field”

L799: remove ‘the’

L808: Replace ‘they’ with the people you are referring to, e.g. local communities

Conclusion: consider redefining the 3 Rs here, for readers that focus on the conclusion section

L866: insert ‘achieved’ or similar, to read ‘Rewetting degraded peatlands is predominately achieved through canal blocking’

L870: include a short justification for involving the community – e.g. to prevent canal blocks being removed by peatland users unaware of restoration goals.

L873: change ‘need to be’ to ‘are’.

Author Response

Dear Reviewer 2,

We thank you for your constructive comments and suggestions to our manuscript. We have edited the manuscript. Please find attached our response to your comments and suggestions. 
